# Estimation of the volatility distribution of organic aerosol combining thermodenuder and isothermal dilution measurements

Evangelos E. Louvaris [1,2], Eleni Karnezi [3], Evangelia Kostenidou [2], Christos Kaltsonoudis [2,3], and Spyros N. Pandis [1,2,3]

[1] Department of Chemical Engineering, University of Patras, Patras, Greece

[2] Institute of Chemical Engineering Sciences, FORTH/ ICEHT, Patras, Greece

[3] Department of Chemical Engineering, Carnegie Mellon University, Pittsburgh, USA

*Correspondence to:* Spyros N. Pandis (spyros@andrew.cmu.edu)

## Abstract

A method is developed following the work of Grieshop et al. (2009) for the determination of the organic aerosol (OA) volatility distribution combining thermodenuder and isothermal dilution measurements. The approach was tested in experiments that were conducted in a smog chamber using organic aerosol (OA) produced during meat charbroiling. A thermodenuder (TD) was operated at temperatures ranging from 25 to 250$^{\circ}$C with a 14 s centerline residence time coupled to a High-Resolution Time-of-Flight Aerosol Mass Spectrometer (HR-ToF-AMS) and a Scanning Mobility Particle Sizer (SMPS). In parallel, a dilution chamber filled with clean air was used to dilute isothermally the aerosol of the larger chamber by approximately a factor of 10. The OA mass fraction remaining was measured as a function of temperature in the TD and as a function of time in the isothermal dilution chamber. These two sets of measurements were used together to estimate the volatility distribution of the OA and its effective vaporization enthalpy and accommodation coefficient. In the isothermal dilution experiments approximately 20% of the OA evaporated within 15 min. Almost all the OA evaporated in the TD at approximately 200$^{\circ}$C. The resulting volatility distributions suggested that around 60-75% of the cooking OA (COA) at concentrations around 500 µg m$^{-3}$ consisted of low volatility organic compounds (LVOCs), 20-30% of semi-volatile organic compounds (SVOCs) and around 10% of intermediate volatility organic compounds (IVOCs). The estimated effective vaporization enthalpy of COA was 100 ± 20 kJ mol$^{-1}$ and the effective accommodation coefficient was 0.06-0.07. Addition of the dilution measurements to the TD data results in a lower uncertainty of the estimated vaporization enthalpy as well as the SVOC content of the OA.

## 1. Introduction

Atmospheric aerosols have a significant impact both on human health (Pope et al., 2009; Caiazzo et al., 2013) and on Earth's climate due to their ability to scatter and absorb solar radiation and their effects on cloud properties and lifetimes (IPCC, 2014). These particles consist of a wide variety of chemical compounds, with organic components representing 20-90% of their submicron mass (Zhang et al., 2007). Organic aerosol (OA) can be emitted directly as primary particles (POA) from various anthropogenic and natural sources or can be formed when gas-phase oxidation products of volatile (VOCs), intermediate volatility (IVOCs) and semi-volatile (SVOCs) organic compounds condense onto pre-existing particles forming secondary organic aerosol (SOA). There is limited knowledge of the sources, chemical evolution, and physical properties of OA due to the complexity of the mostly unknown thousands of constituents of OA. These uncertainties often lead to erroneous predictions of OA concentrations by chemical transport models.

Volatility is one of the most important physical properties of OA as it determines the partitioning of its components between the gas and particulate phases, and eventually their atmospheric fate (Donahue et al., 2012). One of the most common techniques to constrain indirectly aerosol volatility requires the use of a thermodenuder (TD). The aerosol enters a heated tube where the most volatile components evaporate leaving behind the less volatile species in the particulate phase (Burtscher et al., 2001; Kalberer et al., 2004; Wehner et al., 2002, 2004; An et al., 2007). TDs usually consist of two sections: the heating section where the aerosol evaporation takes place and the denuder/cooling section. This second section often contains activated carbon in order to prevent re-condensation of the evaporated components. The typical result of a TD is the mass fraction remaining (MFR) of the aerosol as function of the TD temperature. The MFR depends on aerosol concentration, size, vaporization enthalpy, and potential mass transfer resistances (Riipinen et al., 2010).

TD measurements of OA volatility have received considerable attention recently, and have been performed both in the field (Huffman et al., 2009; Cappa and Jimenez, 2010; Lee et al., 2010; Louvaris et al., 2017) and in the laboratory (Saleh et al., 2008; Faulhaber et al., 2009; Lee et al., 2011). Riipinen et al. (2010) argued that OA practically never reaches equilibrium in a TD at ambient concentration levels. TD measurements were performed by Lee et al. (2010, 2011) using multiple residence times. These authors argued that use of multiple residence times in the heating section of the TD can help to decouple mass transfer effects from thermodynamics. Similar conclusions were reached  also by Riipinen et al.

(2010) and Cappa (2010). Saleh et al. (2012) used a particle concentrator upstream of a TD in order to achieve higher ambient aerosol loadings so that the system could reach equilibrium. Their results suggested accommodation coefficient values around 0.3 for the ambient aerosol that they examined. Volatility measurements based on longer equilibration timescales were performed for POA from a diesel engine and wood combustion using isothermal dilution

(Grieshop et al., 2009). Cappa and Wilson (2011) studied the evolution of the OA mass spectra from lubricating oil and a-pinene oxidation as the particles were heated. They concluded that there were high mass transfer resistances for the SOA produced from $\alpha$-pinene ozonolysis. Saleh et al. (2013) measured the equilibration timescales for the gas to particle partitioning of SOA formed from $\alpha$-pinene ozonolysis using an accommodation coefficient of

the order of 0.1.

        Grieshop et al. (2009) suggested that combination of thermodenuder and dilution measurements can better constrain the OA volatility over a wide range. Karnezi et al. (2014) proposed an algorithm for the combination of the two types of measurements and the derivation of the optimum volatility distribution of OA and its uncertainty. Kolesar et al.

(2015) combining rapid isothermal dilution with TD measurements argued that the volatility of SOA formed from $\alpha$-pinene ozonolysis is mostly independent of the SOA loading during temperature-induced evaporation.

        Most of the previous studies discussed above determined the OA volatility assuming a-priori values for the OA vaporization enthalpy and accommodation coefficient. Since the TD

results are sensitive to these values large uncertainties were reported for the volatility distributions. The performance of different time scale measurements can, in principle at least, allow the estimation of the volatility distribution together with the vaporization enthalpy and accommodation coefficient with lower uncertainty ranges.

        In this study, we continue the development of the experimental technique of Grieshop

et al. (2009) to constrain the volatility distribution of organic aerosol using TD combined with isothermal dilution measurements using the algorithm of Karnezi et al. (2014). The OA mass fraction remaining is measured as a function of temperature in the thermodenuder and as a function of time in a dilution chamber in parallel. TD measurements are corrected for size- and temperature- dependent losses and the dilution system measurements for size-

dependent losses. These two sets of measurements are then used together with the approach of Karnezi et al. (2014) to estimate the volatility distribution of the OA and its effective enthalpy of vaporization ($\Delta H_{vap}$) and effective accommodation coefficient ($a_m$). Cooking OA is used as an example for the application of the method.

## 2. Experimental description

Smog chamber experiments were conducted in the FORTH smog chamber to constrain the volatility distribution of fresh OA emissions from meat charbroiling. The experimental setup is shown in Fig. 1. A metal bellows pump (model MB 602, Senior Aerospace) was used to transfer cooking emissions to the 10 m$^3$ Teflon chamber. Details for the meat charbroiling and the transferring process can be found in Kaltsonoudis et al. (2016a). A TD (Louvaris et

al., 2017) was placed upstream of a High-Resolution Time-of-Flight Aerosol Mass Spectrometer (HR-ToF-AMS Aerodyne Research Inc.) (Decarlo et al., 2006; Canagaratna et al., 2007) measuring the size-composition of the submicron non-refractory material, and a Scanning Mobility Particle Sizer (SMPS 3936 TSI) measuring the particle size distribution. A dilution Teflon chamber (1 m$^3$) was used for the isothermal dilution. The VOCs and the

dilution ratio (DR) were measured by a PTR-MS (Ionicon Analytic). Isotopically labeled butanol (1-butanol-d9, Sigma) was added to the chamber to assist in the measurement of the dilution ratio.

The SMPS was operated at a sampling flow rate of 1 L min$^{-1}$ and sheath flow rate of 5 L min$^{-1}$ sampling every 3 minutes. The HR-ToF-AMS was sampling every three minutes with

0.1 L min$^{-1}$ and was operated in the higher sensitivity mode (V-mode) (DeCarlo et al., 2006). The PTR-MS was sampling with 0.5 L min$^{-1}$. Details about the PTR-MS operation can be found in Kaltsonoudis et al. (2016b).

The TD was operated at temperatures ranging from 25 to 250$^{\text{o}}$C using 15 temperature steps for about half an hour per step. Sampling from the main chamber was alternated

between bypass and TD every 3 min with computer-controlled valves. The changes in the particle mass concentration and size were measured by both the HR-ToF-AMS and the SMPS resulting in thermograms of the MFR as a function of the TD temperature. The OA MFR was calculated as the ratio of organic mass concentration of a sample passing through the TD at time $t_i$ over the average mass concentration of the ambient samples that passed through the

bypass line at times $t_{i-1}$ and $t_{i+1}$. The sample residence time in the centerline of the TD was 14 s at 298 K corresponding to an average residence time in the TD of 28 s. The temperature profile in our TD (both in the longitudinal and radial directions) has been analysed by Lee et al. (2010). The change in volumetric flowrate due to the change in temperature along the TD is taken into account by the Riipinen et al. (2010) TD model used in this work.

The dilution chamber was initially partially filled with clean air. Then, the metal bellows pump was used to transfer cooking emissions from the main chamber to it, diluting

them in the process to close to ambient concentration levels. The aerosol was transferred from the main to the dilution chamber only once and then its evolution with time was followed. Dilution measurements were performed every 9 min by both the SMPS and the HR-ToF-AMS. The SMPS sampling flow rate was 1 L min$^{-1}$. The dilution ratio was calculated as the ratio of the PTR-MS *m/z 66* concentration of the main chamber over the PTR-MS *m/z 66* concentration of the dilution chamber. The dilution ratios during the isothermal dilution experiments are shown in Table 1 and were 10 ± 0.5 and 14 ± 0.5 for Experiments 1 and 2 respectively. These remained constant during the experiments. Table 1 also summarizes the characteristics of each experiment. The residence time in the dilution chamber was a few hours. The mass fraction as a function of time during isothermal dilution was measured as the ratio of mass concentration at time $t_i$ over the initial mass concentration in the dilution chamber at time $t_0$.

## 2.1 Loss corrections

The thermodenuded OA was corrected for particle losses in the TD. Particle losses were measured as a function of the TD temperature and particle size using sodium chloride particles and the same flow rate as that used in the experiments. The average loss fraction in the 0.1-1 μm size range for each temperature was used for the correction of the AMS results and the accuracy of the correction was tested using the SMPS number distributions and the approach of Lee et al. (2010).

The OA concentrations during the isothermal dilution experiments were also corrected for size dependent wall losses during the experiments. These losses were calculated for each experiment using the number concentration distributions measured by the SMPS. Following Pathak et al. (2007) the wall loss rate constant $k_w(D_p)$ was estimated from least-square fits of the natural logarithm of the SMPS particle number distributions values for each size as a function of time in the latter stages of the experiment when evaporation was negligible. Fig. S1 shows these loss rate constants as function of particle size for Experiment 1. These size-dependent loss corrections were applied to the measured number distribution at each time step allowing the estimation of the corrected number and volume distributions. The accuracy of the corrections can be evaluated using the temporal evolution of the corrected total number concentration in the chamber. Given the low number concentrations (around 5000 particles cm$^{-3}$) coagulation is negligible and the corrected total number concentration should be approximately constant. The corrected number concentration varied by less than 10% during the three hours of the experiment (Fig. S2a). Even if the measured mass concentration was

reduced by approximately 50% in the dilution chamber, the evaporation resulted in only 20%

mass reduction (Fig. S2b). The other 30% was due to losses of particles to the walls of the

dilution chamber. The change in the aerosol size distribution is consistent with the above

result. In Exp. 1 the number mode diameter decreased from 93 nm to 87 nm corresponding to

an 18% reduction in volume.

Similar results were obtained also during Experiment 2. The corrected number

concentration varied by less than 5% (Fig. S3a), suggesting that the correction was quite

accurate. The evaporation resulted in a 20% mass reduction as shown in Fig. S3b. The

shifting of aerosol size distributions to smaller sizes was once more obvious. The aerosol

number mode diameter decreased from 115 to 105 nm suggesting a 24% reduction in volume.

As another quality assurance test the mass fraction values measured by the SMPS were

compared to those measured by the HR-ToF-AMS assuming that the collection efficiency of

the latter remained constant as the OA evaporated (Fig. S4). The corresponding differences

between the two measurements were a few percent or less.

**2.2 Determination of the volatility distributions**

The dynamic mass transfer model of Riipinen et al. (2010) together with the error

minimization approach proposed by Karnezi et al. (2014) were used for the determination of

the volatility distributions. Inputs for the model included the initial OA mass concentrations

for the TD and the isothermal dilution chamber obtained by the HR-ToF-AMS, the initial

particle sizes obtained by the SMPS, the residence times of both systems, and the dilution

ratio of the isothermal dilution system. The initial mass concentration for the TD experiments

was of the order of $10^2$ μg m$^{-3}$. Table 1 summarizes these inputs of the model. The corrected

mass fraction values determined by the HR-ToF- AMS were used as system inputs for the

calculations.

The volatility distribution in the volatility basis set framework is expressed with a range

of logarithmically spaced $C*$ bins along a volatility axis (Donahue et al., 2006). For our

analysis a set of six volatility bins ranging from $10^{-3}$ to $10^3$ μg m$^{-3}$ were used.


## 3. Results and discussion

### 3.1 Volatility distribution of cooking organic aerosol (COA)

Applying the approach of Karnezi et al. (2014) the volatility distribution of the COA was estimated. The distribution for Experiment 1 is depicted in Fig. 2. The average volatility, defined as the average $\log_{10}C^*$ weighted by the mass fraction of each bin was approximately 0.1 µg m$^{-3}$. This average volatility is a useful metric for the comparison of volatility distributions when the same volatility range is used. According to these results the COA at around 550 µg m$^{-3}$ consisted of 60% low volatility organic compounds (LVOCs), 30% semi-volatile (SVOCs), and 10% intermediate volatility (IVOCs) organic compounds (Fig. 3c). The estimated effective vaporization enthalpy was $100 \pm 14$ kJ mol$^{-1}$ (Fig. 3a) and the effective accommodation coefficient was equal to 0.06 but with corresponding uncertainty range covering more than an order of magnitude (Fig. 3b). The corresponding TD thermogram and the dilution curve for Experiment 1 are depicted in Fig. 4. Almost all the COA evaporated at 200$^{o}$C, while approximately 20% of the COA evaporated at ambient temperature after isothermal dilution. The model reproduced pretty well the measured MFR by the TD but tended to overpredict the measured evaporation during dilution. The model estimated that 25% of the COA evaporated at ambient temperature with the concentration decreasing to 19 µg m$^{-3}$ instead of the observed 21 µg m$^{-3}$. According to the model the small amount of IVOCs that existed initially in the particle phase evaporated at 50$^{o}$C in the TD. The SVOCs evaporated at 125$^{o}$C and the COA remaining at higher temperatures consisted entirely of LVOCs (Fig. 4). The IVOCs, according once more to the model, evaporated after 10 min of dilution and the SVOCs after approximately 30 min. The relatively prompt evaporation during dilution is interpreted by the model as evidence that any resistances to mass transfer in this system were modest. The fact that only 20% of the COA evaporated during these first few minutes, suggests that the contribution of the more volatile OA components (IVOCs and part of the SVOCs) were also modest. After this 20% of the evaporated the system reached equilibrium and evaporation stopped. Table S1 summarizes the estimated volatility distribution along with the estimated effective parameters that affect volatility and the calculated average volatilities for the two experiments.

Kostenidou et al. (2009) proposed the theta angle (θ) as an indicator of mass spectra similarity by treating AMS spectra as vectors and calculating the corresponding angle θ. Lower θ imply more similar spectra. Fig. 5a compares the average HR-ToF-AMS normalized mass spectra of the COA at ambient temperature (25$^{o}$C) to the average normalized spectra in

the TD at 200$^\circ$C. The two spectra were calculated by averaging the corresponding measurements during the experiment. There was very little temporal variation of either the ambient temperature or thermodenuded spectra (theta angles less than 2 degrees). The 25$^\circ$C and 200$^\circ$C spectra were quite similar to each other having an angle θ of 11 degrees (R$^2$=0.958). This suggests that the least volatile COA components were quite similar to the

total COA from the AMS' point of view. The AMS does examine mainly the small fragments of the corresponding compounds therefore the volatility differences in this case may be due mainly to the size of the molecules and not so much to their chemical characteristics (e.g., acids versus olefins). Fig. 5b depicts the comparison of HR-ToF-AMS mass spectra at the onset of dilution with that one hour later, and the one at the end of the experiment. The

resulting θ angles between the compared mass spectra were 3 to 4 degrees (R$^2$ ranging from 0.994 to 0.997) showing the similarity of the average OA composition during the dilution experiment.

The estimated volatility distribution of Experiment 2 is shown in Fig. 2. The average $\log_{10}C^*$ was 0.05 μg m$^{-3}$. The COA consisted of 75% LVOCs and 25% of SVOCs (Fig. 3c).

The vaporization enthalpy was 85 ± 9 kJ mol$^{-1}$ (Fig. 3a) and the accommodation coefficient was equal to 0.07 (Fig. 3b). The differences especially in the evaporation at the lower temperatures between the two experiments can be due to the differences in the COA produced in the two experiments. However, the θ angle for the two COA spectra in the two experiments was only 5$^\circ$.

The thermogram of TD measurements and the corresponding dilution curve of Experiment 2 are depicted in Fig. 6. Almost all the COA evaporated at 225$^\circ$C in the TD. 20% (from 7.5 to 6 μg m$^{-3}$) of the COA evaporated at ambient temperature during dilution. The model reproduced quite well the corresponding TD measurements below 75$^\circ$C but tended to overpredict the observed evaporation at higher temperatures. At the same time the model

tended to slightly underpredict the observed evaporation at room temperature. According to the model the SVOCs evaporated at 130$^\circ$C and the COA remaining at higher temperatures consisted entirely of LVOCs (Fig. 6c). During isothermal dilution the model predicted that after 10 minutes the SVOCs evaporated (Fig. 6d).

The TD measurements even at 250$^\circ$C could be reproduced assuming that least volatile

COA components had a C$^*$=10$^{-3}$ μg m$^{-3}$. However, the existence of components with even lower volatility (extremely low volatility organic compounds, ELVOCs) cannot be eliminated. Around 5% of the COA did not evaporate even at the highest temperature used in

the system. Nie et al. (2017) have provided evidence about the connection of ambient ELVOCs with humic like substances (HULIS).


## 3.2 Sensitivity analysis

The sensitivity of the above results to the parameter estimation approach was investigated. Two types of sensitivity tests were conducted: one by assuming specific effective accommodation coefficients and estimating the volatility distributions and effective

vaporization enthalpies and one by assuming different effective vaporization enthalpies and estimating the volatility distributions and accommodation coefficients using the approach of Karnezi et al. (2014).

### 3.2.1 Sensitivity to the accommodation coefficient

During these tests the volatility distributions and effective vaporization enthalpies were estimated for both experiments while assuming fixed accommodation coefficient values. Table S2 and Fig. S5 summarize the estimated volatility distributions during these tests. Assuming an accommodation coefficient of 0.01 which is approximately half an order of magnitude lower than the estimated one for the base case (0.01 instead of 0.06 for

Experiment 1 and 0.07 for Experiment 2), the IVOC fraction remained the same but the SVOCs increased by 15% and the LVOCs decreased by the same amount compared to the base case results. Assuming an $a_m$ equal to 0.1 the LVOCs increased by around 10% and the SVOCs decreased by the same amount, while the IVOCs remained once again the same (Fig. S5). For a further increase of $a_m$ to unity the LVOCs increased by 15% and the SVOCs

decreased by the same amount, while the IVOC fraction remained the same (Fig. S5). The estimated effective vaporization enthalpies were almost the same as those estimated in the base case (around 100 and 85 kJ mol$^{-1}$ for Experiment 1 and Experiment 2 respectively) for all the investigated accommodation coefficients during this analysis (Table S2).

The predicted thermograms of the TD measurements and the corresponding predicted

dilution curves of both experiments are shown in Fig. S6. For Experiment 1 assuming an accommodation coefficient equal to 0.01 the model performance deteriorated slightly at the low (25-80$^o$C) and high (180-220$^o$C) temperatures but improved in the middle (120-150$^o$C). The cases of $a_m$=0.1 and 1.0 the predicted thermograms were almost the same with the one predicted in the base case (Fig S6a). The predicted dilution curve using an accommodation

coefficient of 0.01 reproduced a little better the observed mass fraction values during the first hour of isothermal dilution compared to the base case. For an accommodation coefficient equal to unity the evaporation at ambient temperature was overestimated during the first hour of the experiment and for accommodation coefficient 0.1 the dilution curve was almost the same with that predicted for the base case (Fig. S6b). The predicted thermograms for

Experiment 2 were quite similar for all accommodation coefficient examined (Fig. S6c). With the exception of the $a_m$=0.01 the other three simulations (for $a_m$= 0.07, 0.1 and 1.0) reproduced the dilution observations quite well (Fig. S6d).

Summarizing, varying the accommodation coefficient from 0.01 to 1.0 compared to the estimated 0.06-0.07 resulted in negligible changes in the estimated enthalpy of vaporization

and the IVOC content of COA. The SVOCs and LVOCs changed by less than 15% in these tests. The deterioration in the performance of the model was small, underlying the difficulty of obtaining accurate values of the effective accommodation coefficient from such measurements in complex systems.

**3.2.2 Sensitivity tests to vaporization enthalpies**

During these tests the volatility distributions and effective accommodation coefficients were estimated for both experiments assuming values of the effective vaporization enthalpy. Values of 120 and 60 kJ mol$^{-1}$ were used for both experiments to test the sensitivity of our results to $\Delta H_{vap}$. These should be compared to the estimated values of 100 kJ mol$^{-1}$ for the

first experiment and 85 kJ mol$^{-1}$ for the second.

For the high value of the vaporization enthalpy (120 kJ mol$^{-1}$) the estimated volatilities were lower by approximately half an order of magnitude compared to that of the base case (Table S2) for both experiments. The LVOCs increased by 5-10% and the SVOCs decreased by the same amount, while the IVOC fraction remained approximately the same. Fig. S7

shows the estimated volatility distributions and COA compositions of both experiments for all the cases of this analysis. The estimated accommodation coefficients were almost half an order of magnitude lower compared to that of the base case values (Table S2).

Assuming a vaporization enthalpy of 60 kJ mol$^{-1}$ the corresponding volatilities for both experiments increased by approximately a factor of 2. The LVOC fraction for this case

decreased by 5-10% and a corresponding increase was estimated for the SVOCs. Once again the IVOC fraction remained the same. The estimated accommodation coefficients were similar to the base case.

For both experiments the increase of the vaporization enthalpy resulted in an overprediction of the evaporation in the TD failing to reproduce the results of Experiment 1. On the other hand the corresponding decrease in vaporization enthalpy led to the opposite problem. The changes in the model's ability to reproduce the dilution measurements were, as expected, less sensitive to $\Delta H_{vap}$.

The above results suggest that changes in vaporization enthalpy by 15-40 kJ mol$^{-1}$ produce changes in the volatility distribution by less than half an order of magnitude. Higher values of the enthalpy are balanced with lower volatilities and vice versa. The accommodation coefficient is more sensitive in this case to higher vaporization enthalpy values than to lower ones.

## 4. Benefits of combining TD and isothermal dilution

In order to evaluate the benefits of the combination of thermodenuder and isothermal dilution measurements the above results were compared to the results obtained by using only the thermodenuder data. The algorithm of Karnezi et al. (2014) was used once more to estimate the volatility distributions, the vaporization enthalpy, and accommodation coefficient based only on the thermodenuder measurements.

The corresponding results are shown in Fig. 7 and Table S1. The combination of thermodenuder and dilution measurements resulted in a less volatile COA in both cases. In Experiment 1 the average volatility was reduced by almost half an order of magnitude (from 0.44 to 0.1 μg m$^{-3}$) due to the inclusion of dilution data. For Experiment 2 the corresponding reduction was approximately a factor of 2. The combined approach suggests that COA consisted of 60% LVOCs, 30% SVOCs, and 10% IVOCs while the TD-only approach results in almost 40% LVOCs, 50% SVOCs, and 10% IVOCs for Experiment 1. For Experiment 2 the combined approach once again suggested that the COA consisted of 77% LVOCs and 23% SVOCs, while the TD-only approach suggested of 68% LVOCs and 32% SVOCs.

In both experiments the use of only the thermodenuder measurements resulted in an overestimation of the SVOCs by 10-20% and a subsequent reduction by the same amount of the LVOCs. The combination of thermodenuder and isothermal dilution measurements led to a reduction of the uncertainty range for the more volatile OA components with effective saturation concentrations from 10 to 1000 μg m$^{-3}$. The uncertainty ranges of the estimated effective vaporization enthalpies were reduced from 15-20 kJ mol$^{-1}$ to 10-15 kJ mol$^{-1}$ when

the dilution data were included in the analysis. There was little change in the uncertainty of
the accommodation coefficients.

## 5. Conclusions

The approach of Grieshop et al. (2009) for the determination of the organic aerosol (OA) volatility distribution combining thermodenuder and isothermal dilution measurements is extended and combined with the optimization algorithm proposed by Karnezi et al. (2014).
The combination of TD and isothermal dilution for the estimation of the volatility distribution was tested for cooking OA from meat grilling. Size dependent losses were taken into account for the correction of both thermodenuder and dilution measurements.

All the COA evaporated in the TD at 225$^{o}$C while 80% remained after dilution by a factor of 10 at ambient temperature. The COA average volatility was between 0.05 and 0.1
μg m$^{-3}$. The COA at around 500 μg m$^{-3}$ consisted of 60-75% LVOCs, 25-30% SVOCs, and a small fraction (10%) of IVOCs. The estimated effective vaporization enthalpy was 100 ± 15 kJ mol$^{-1}$, and the effective accommodation coefficient was 0.06-0.07 with corresponding uncertainty range of one order of magnitude. These values should be applicable to COA produced during pork meat charbroiling.

Changes of the accommodation coefficient of half an order of magnitude result in similar magnitude changes of the average volatility. The estimated vaporization enthalpy was almost the same with the reported value of the base case. Similar results were found for a change of the effective vaporization enthalpy by 15-40 kJ mol$^{-1}$. The COA composition exhibited changes in the LVOC and SVOC fractions by 5-15% while the IVOCs remained
practically the same.

The use of only TD measurements resulted in an overestimation of the SVOC fraction of COA leading to a shifting of volatility towards higher values. Combination of TD and dilution results in a lower uncertainty of the estimated effective vaporization enthalpy. The dilution measurements also help constrain the contribution of the more volatile OA
components (SVOCs) to the total OA concentration. On the other hand, the volatility distribution of the LVOCs is based mainly on the TD data.


**Acknowledgements**

This research was supported by the US National Science Foundation (Award number 1455244) and the Greek "ARISTEIA" action of the "Operational Programme Education and Lifelong Learning" co-funded by the European Social Fund (ESF) and National Resources.

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

510

**Table 1:** Summary of the experimental conditions

| Experiment | Initial concentration ($\mu g\ m^{-3}$) | | Initial volume mode diameter (nm) | | Average dilution ratio |
| --- | --- | --- | --- | --- | --- |
| | Main chamber | Dilution chamber | Main chamber | Dilution chamber | Dilution chamber |
| 1 | 541 | 26.5 | 248 | 210 | $11 \pm 0.5$ |
| 2 | 632 | 7.4 | 284 | 218 | $14 \pm 0.5$ |

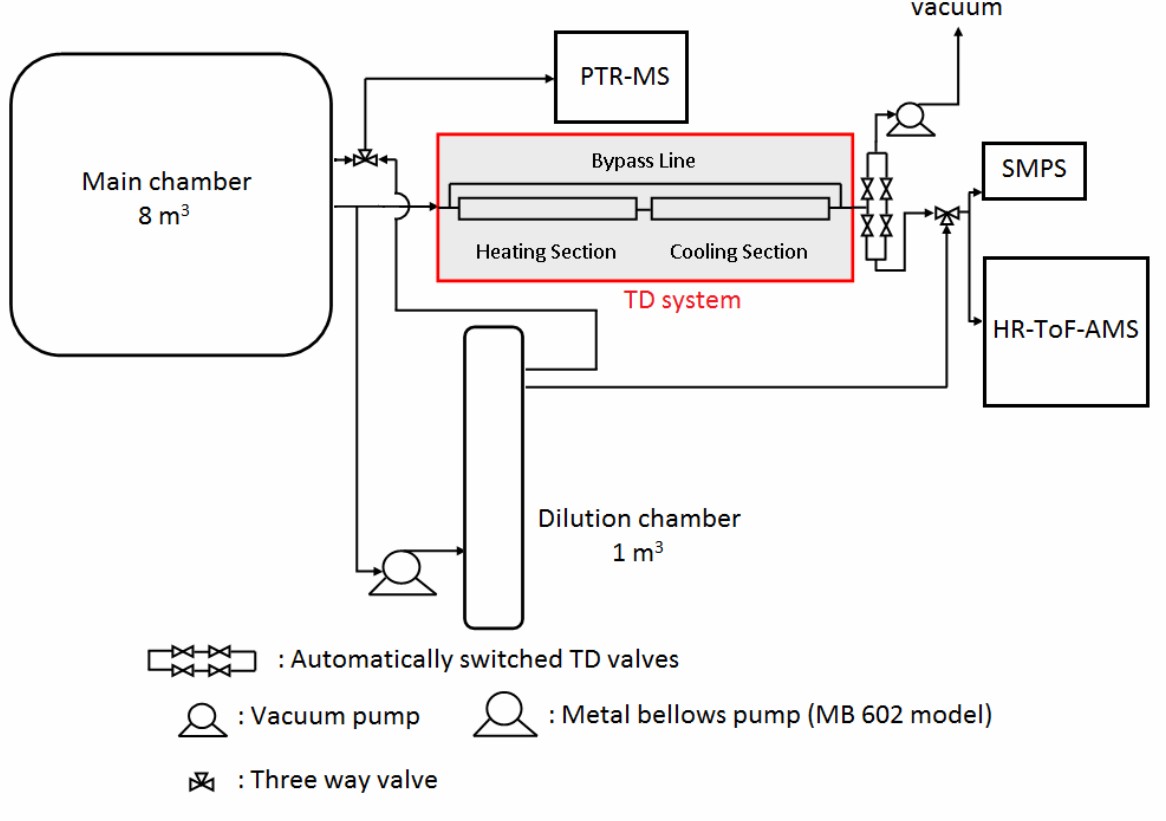

**Figure 1:** Schematic of the experimental setup used in the experiments. The COA in the main chamber was characterized using a TD a HR-ToF-AMS, and an SMPS. A metal bellows pump was used to transfer OA from the main to the dilution chamber. The COA in the dilution chamber was measured by the HR-ToF-AMS and the SMPS. A PTR-MS was used to measure the dilution ratio.

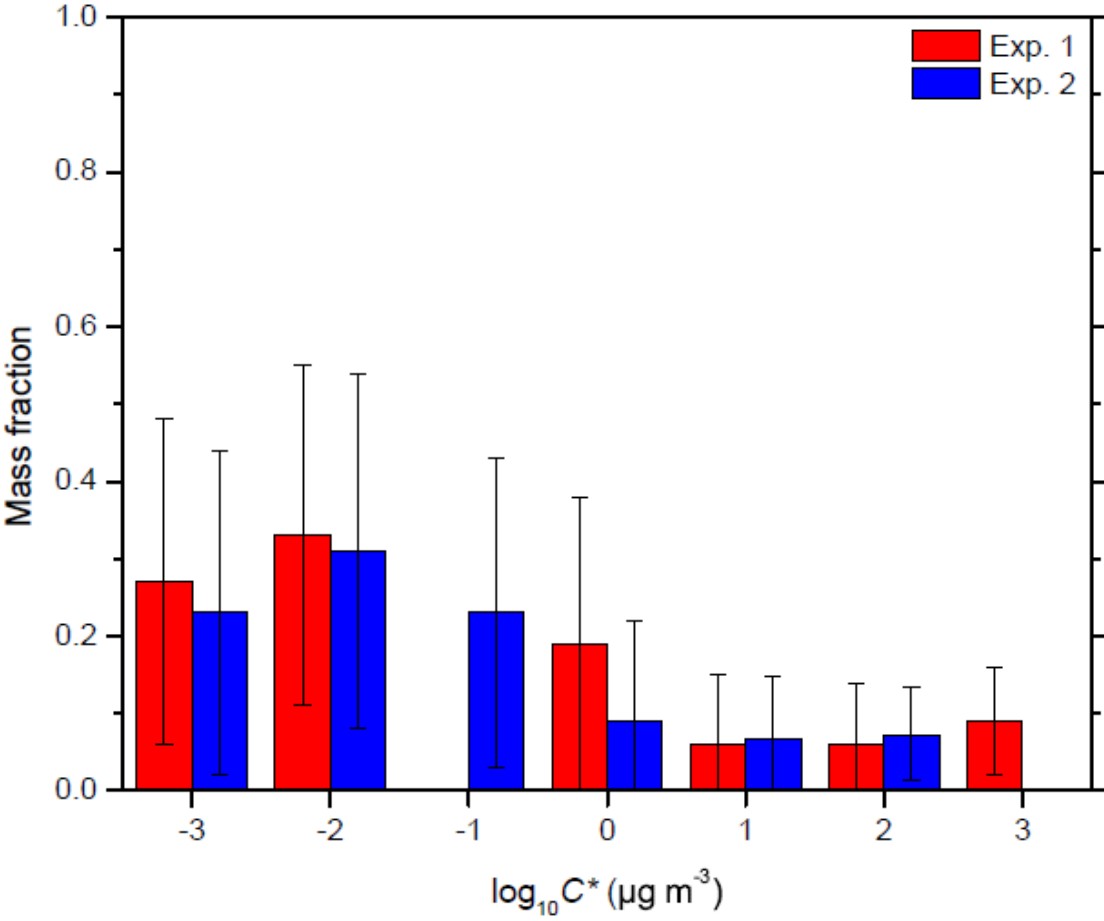

**Figure 2:** Estimated volatility distributions for Experiment 1 (red bars) and for Experiment 2 (blue bars) using the approach of Karnezi et al. (2014). The error bars represent the uncertainty of the estimated mass fractions.

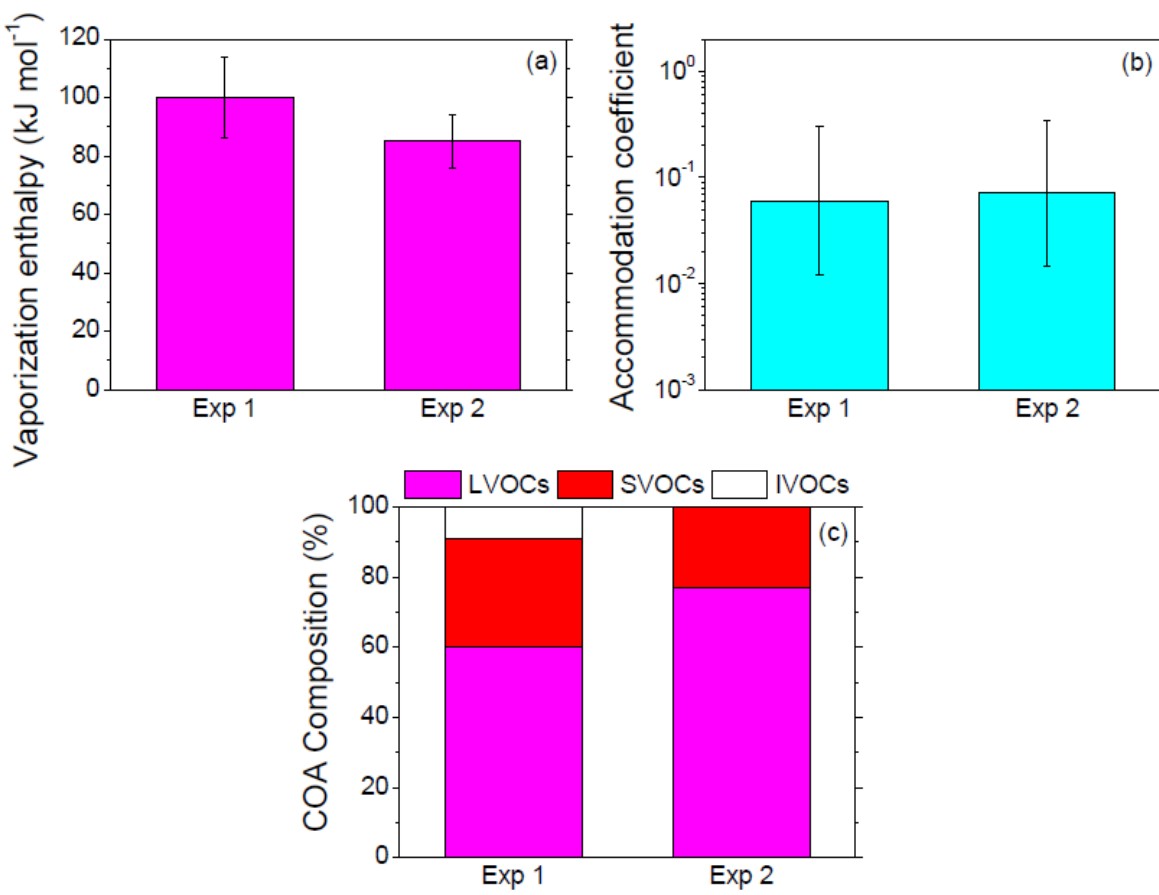

**Figure 3:** (a) Estimated effective vaporization enthalpies along with their uncertainties for both experiments using the approach of Karnezi et al. (2014). (b) Estimated effective accommodation coefficients along with their uncertainties for both experiments. (c) COA mass composition of both experiments. LVOCs are represented in magenta, SVOCs in red, and IVOCs in white.

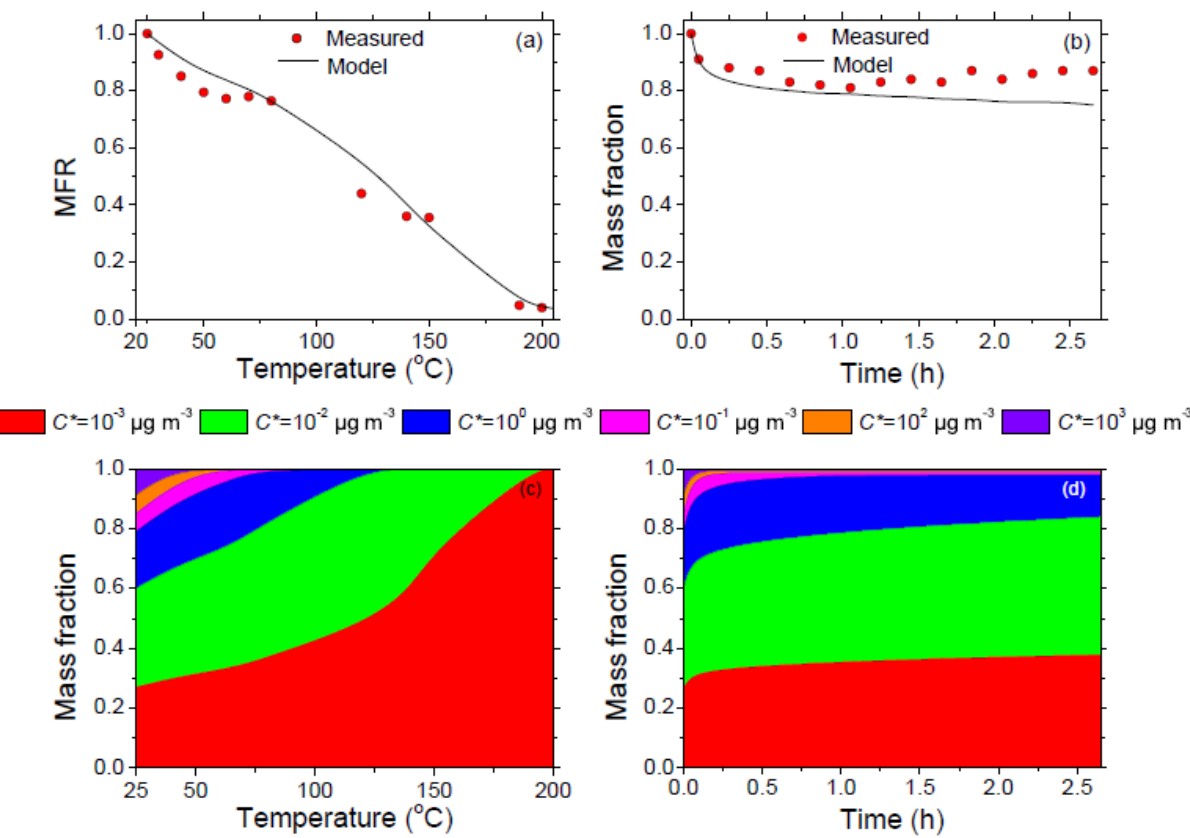

**Figure 4:** (a) Thermogram of the OA TD measurements of Experiment 1. Red circles represent the loss corrected measurements and the black line represents the best fit estimated by the model of Karnezi et al. (2014) (b) Mass fraction during isothermal dilution as a function of time of Experiment 1. Red circles represent the loss corrected measurements and the black line the estimated best model fit. (c) COA mass fraction for different effective saturation concentrations as a function of TD temperature. Red color represents the contribution of the effective saturation concentration $C* = 10^{-3}$ µg m$^{-3}$, green the contribution of the $C* = 10^{-2}$ µg m$^{-3}$, blue the $C* = 1$ µg m$^{-3}$, magenta the $C* = 10$ µg m$^{-3}$, orange the $C* = 10^{2}$ µg m$^{-3}$, and violet the $C* = 10^{3}$ µg m$^{-3}$. (d) COA mass fraction for different effective saturation concentrations as a function of time during isothermal dilution.

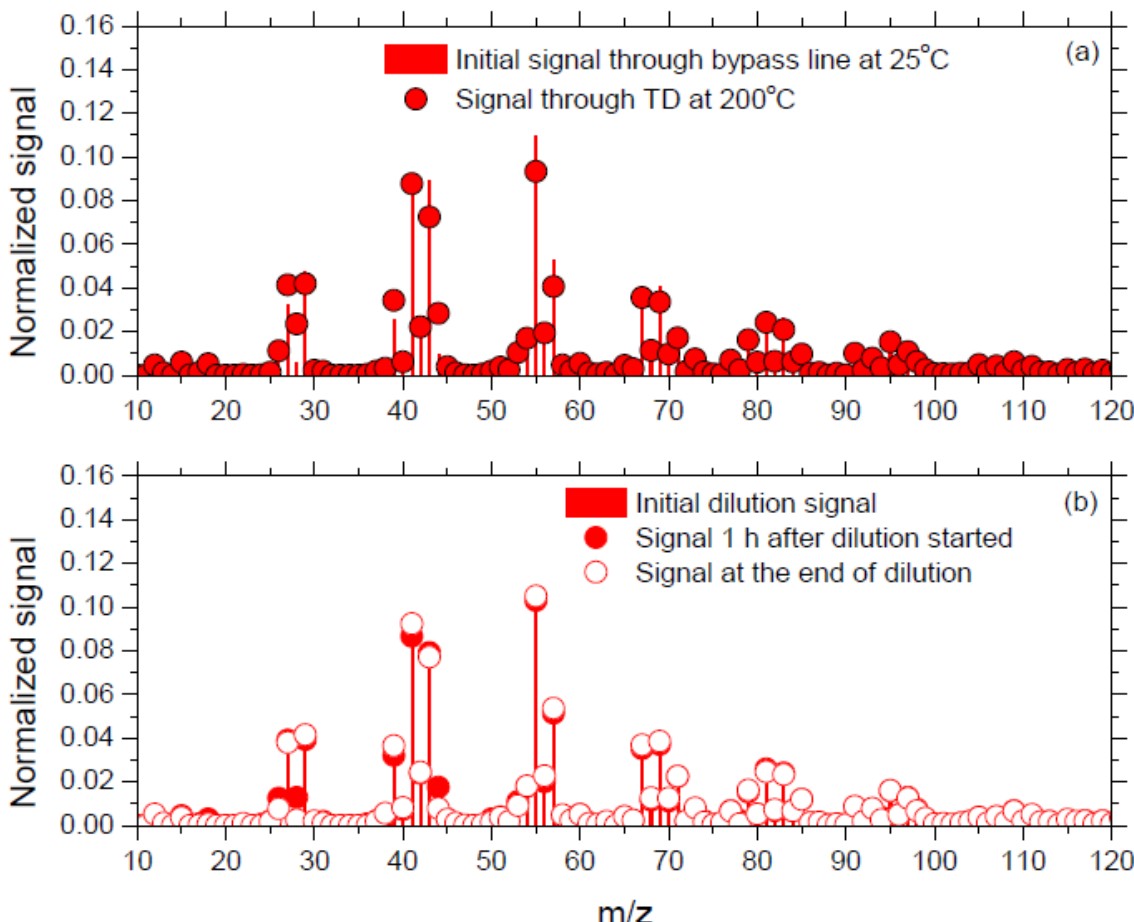

**Figure 5:** (a) Normalized HR-ToF-AMS mass spectra of the initial measurements at ambient temperature through the bypass line (red bars) compared to those measured in the TD at 200°C (red circles). (b) Normalized HR-ToF-AMS mass spectra at the onset of dilution experiment (red bars) compared to those measured after one hour (solid circles), and to those measured at the end of the experiment (open circles).

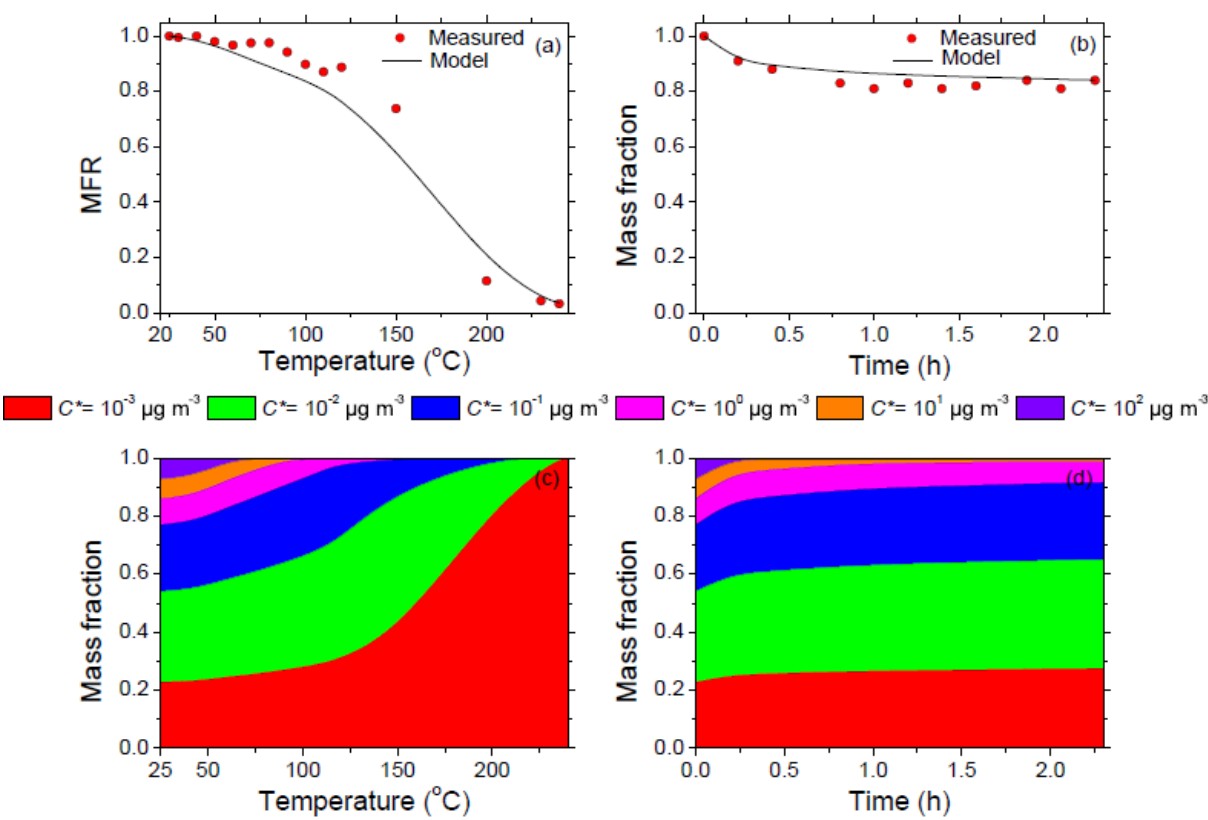

**Figure 6:** (a) Thermogram of the OA TD measurements of Experiment 2. Red circles represent the loss corrected measurements and the black line represents the best fit estimated by the model of Karnezi et al. (2014). (b) Mass fraction during isothermal dilution as a function of time of Experiment 2. Red circles represent the loss corrected measurements and the black line the estimated best model fit. (c) COA mass fraction for different effective saturation concentrations as a function of TD temperature. Red color represents the contribution of the effective saturation concentration $C^* = 10^{-3}$ µg m$^{-3}$, green the contribution of the $C^* = 10^{-2}$ µg m$^{-3}$, blue the $C^* = 10^{-1}$ µg m$^{-3}$, magenta the $C^* = 1$ µg m$^{-3}$, orange the $C^* = 10$ µg m$^{-3}$, and violet the $C^* = 10^{2}$ µg m$^{-3}$. (d) COA mass fraction for different effective saturation concentrations as a function of time during isothermal dilution.

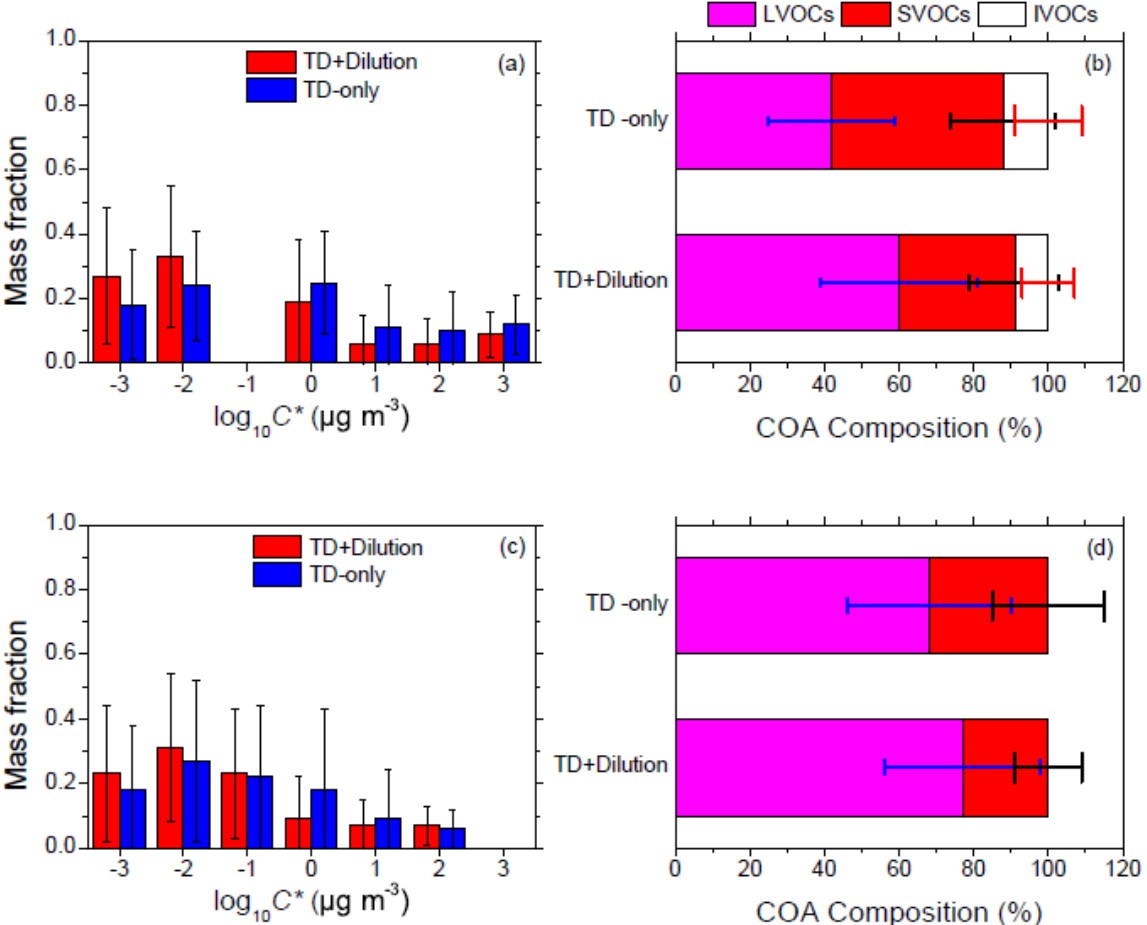

**Figure 7:** (a) Estimated volatility distributions of the COA along with their uncertainties of Experiment 1 using the approach of Karnezi et al. (2014). Red bars represent the volatility distribution using the combination of thermodenuder and isothermal dilution measurements whereas blue bars represent the volatility distribution using only thermodenuder measurements (b) Corresponding COA compositions for the two cases of (a) along with their corresponding uncertainties (±1 standard deviation). LVOCs are in magenta, SVOCs in red, and IVOCSs in white. The uncertainty range of IVOCs, SVOCs and LVOCs are shown in red, black and blue respectively. (c) Estimated volatility distributions of the COA of Experiment 2 using the approach of Karnezi et al. (2014). Red bars represent the volatility distribution using the combination of thermodenuder and isothermal dilution measurements whereas blue bars represent the volatility distribution using only thermodenuder measurements. (d) Corresponding COA compositions for the two cases of (c) along with their corresponding uncertainties (±1 standard deviation). LVOCs are in magenta and SVOCs in red. The uncertainty ranges of SVOCs and LVOCs are shown in black, and blue respectively.