# Peer review of "Estimation of the volatility distribution of organic aerosol combining thermodenuder and isothermal dilution measurements"

_Atmospheric Measurement Techniques, 2017_

## Referee Comment (RC1) · Anonymous Referee #1 · 13 Jun 2017

The authors report on two experiments from a "new experimental method" that is really a combination of two existing methods: a thermodenuder and a dilution chamber. They use the model of Karnezi et al. (2014) to analyze the observations and extract volatility distributions for organic aerosol generated from meat charbroiling. The current study adds an explicit experimental element that was absent from the theoretical Karnezi et al. study. Given that the primary difference between this study and the Karnezi et al. study is the experimental measurement, I think that the manuscript requires much more detail regarding the experimental setup and what makes this new. Also, I think that the authors need to do a more complete job with data quality assurance, or at least need to convince me that their dilution measurements are correct. Finally, I think that

the authors need to give more appropriate credit to work that has come before this. With substantial revision, I think that this paper could be acceptable for publication.

Specific comments follow below.

In their introduction, Louvaris et al. completely downplay the similarity of their work to the previous work of Grieshop et al. (2009, ES&T). The title if the Grieshop paper is "Constraining the Volatility Distribution and Gas-Particle Partitioning of Combustion Aerosols Using Isothermal Dilution and Thermodenuder Measurements." This is extremely similar to the title of the current paper: "Estimation of the volatility distribution of organic aerosol combining thermodenuder and isothermal dilution measurements." In the introduction, the authors mention that "Volatility measurements [by Grieshop] based on longer equilibration timescales were performed for POA from a diesel engine and wood combustion using isothermal dilution." But, they do not even mention that Grieshop also performed thermodenuder measurements. This seems to me like intentional obfuscation, especially since the authors also note that "Karnezi et al. (2014) proposed that volatility distribution of complex OA can be better constrained combining TD and isothermal dilution measurements," and later "we develop a new experimental technique to constrain the volatility distribution of organic aerosol using TD combined with isothermal dilution measurements following the suggestions of Karnezi et al. (2014)." Why mention Karnezi but not Greishop in this context? Simply because one of the authors here is a co-author on the Karnezi paper but not the (earlier) Greishop paper? I am concerned about this clear, seemingly intentional downplaying of previous measurements and methods.

The dilution experiments are insufficiently described. Was aerosol transferred once to the dilution chamber and then air sampled from it while the bag was allowed to collapse? Or was make up air continuously added? What was the flow rate through/out of the bag?

Comparison of Fig. S3 and S4 indicates that for Experiment 2 the "corrected" number

loss is ~10-15% and, importantly, is very similar to the "corrected" mass loss. In other words, comparison between these figures suggests, at least to me, that the mass loss is entirely driven by number loss for this experiment. I have similar concerns regarding the dilution data for experiment 1, from comparison between Fig. S2 and Fig. S6. Put another way, it seems that the number-normalized mass loss for the dilution experiments is close to zero (MFR close to 1). To be convincing that the mass loss is real, I think that the authors need to consider the extent to which the particle size distribution shifted. Was shrinkage observed to an extent that is consistent with 10-15% mass loss? Currently, I do not find the "corrected" mass loss experimental observations convincing. Consequently, I have concerns over the resulting model interpretation of the observations and the entire paper.

L109: Since the authors are looking at "fresh" OA, it is unclear why OH radical concentrations are mentioned. D9 butanol is simply used as a dilution tracer.

L206 and Fig. 5: The experiments took nearly 8 h (30 mins at each of 15 temperatures). I think that the authors need to provide more information regarding exactly when the various spectra were measured. Is the "ambient" spectrum an average over those ~8 h? Is it the average of the two measurements that came before/after the 200 C measurement? How similar in time was the ambient spectrum to the 200 C spectrum? Related, how similar were the ambient spectra measured at the beginning of the measurement versus 8 h later at the end of the measurement? Have things evolved over time due to the particles being suspended in the chamber? Granted, the spectra are quite similar, as the authors note, but more detail is required. Further, the authors simply note that the spectra are similar. But, their calculations suggest that, perhaps, they should be different because the particle composition at ambient vs. 200 C is different. This should be discussed.

L217: The authors note that there are "small" differences between the two experiments. But I would argue that the differences, especially in the TD experiments, are actually quite large. I have included the figures from the paper below so that they can be

directly compared. It is clear that the behavior at lower temperatures is dramatically different. One experiment indicates almost no evaporation until T > 80 C while the other indicates substantial (20% loss) evaporation at temperatures just over ambient. Further discussion is required.

L307 and elsewhere: The authors report the "average" volatility in a number of locations. First, it is not clear how this is calculated. Is it a linear average? A logarithmic average? Second, in reporting this number the authors seem to be making an a priori assumption that this is a meaningful number. This is not a value that is commonly reported. What is this mean meant to represent and how is it useful? I could have two very different distributions, for example one bimodal and one monomodal, that have the same average. These would exhibit very different behavior though.

Experiment #1   Experiment #2

[Figure]

**Fig. 1.**

---

## Referee Comment (RC2) · Anonymous Referee #2 · 3 Jul 2017

This paper provided a new technique to measure/estimate the volatility distribution of organic aerosols by combining a thermodenuder (TD) and a dilution system. Cooking OA was used as an example to show the performance of this technique. In general, it could be a better technique than a separated TD system, which would be suffered by the possible processes occurred during the heating, e.g. thermal decomposition. The manuscript is overall well written and the topic fits the scope of AMT. I therefore recommend this manuscript can be published after some revision.

1. More discussions/statements are needed to explain how these two systems are combined. What is the role of dilution system in estimating the volatility distribution of

targeted aerosol?

2. How to understand the differences between the modelled and experimented data during isothermal dilution (Fig. 4b)? It looks that the evaporation only occurred during the first few minutes, but not a continuous evaporation. Will this influence the understanding and estimation of the volatility distribution?

3. It is highly possible that COA contain some ELVOC, e.g. HULIS (Nie et al., 2017@ACP). I suggest the author to provide a more bin of ELVOCs?

4. How to change the dilution time, by changing flow?

5. Why the mass spectrums look so similar before and after heating or dilution (25°C vs 200 °C)? The volatility of organics should be correlated to their oxidation state (e.g. O/C), molecular weight et al. (E)LOVC tends to have a higher oxidation state and large molecular weight. This indicate there should be some differences for the mass fragmentation between evaporated mass and remained mass. PMF could be used for the AMS measurements in case of several heating steps, to make the change of mass spectrums clearer, e.g. Hong et al., 2017@ACP..

6. Any ideas about the phase state of produced COA, which could influence the volatility measurement.

7. The residence time is effective residence time or total residence time? There would be a temperature profile in the thermodenuder? A effective residence time should be carefully considered.

8. The given vaporization enthalpy of COA in this study is a universal value, or can only be used in this work?

---

## Author Comment (AC1) · 6 Aug 2017

**(1)** *The authors report on two experiments from a "new experimental method" that is really a combination of two existing methods: a thermodenuder and a dilution chamber. They use the model of Karnezi et al. (2014) to analyze the observations and extract volatility distributions for organic aerosol generated from meat charbroiling. The current study adds an explicit experimental element that was absent from the theoretical Karnezi et al. study. Given that the primary difference between this study and the Karnezi et al. study is the experimental measurement, I think that the manuscript requires much more detail regarding the experimental setup and what makes this new.*

*Also, I think that the authors need to do a more complete job with data quality assurance, or at least need to convince me that their dilution measurements are correct. Finally, I think that the authors need to give more appropriate credit to work that has come before this. With substantial revision, I think that this paper could be acceptable for publication. Specific comments follow below.*

We do appreciate the constructive comments and suggestions by the referee. We have done our best to address all of them and to improve the manuscript accordingly. Indeed, the present work is an effort to apply in the laboratory the approach suggested by the theoretical analysis of Karnezi et al. (2014). As expected, an experiment looks always a lot easier on paper (or as the output of a code) than in practice. The present work focuses on the experimental uncertainties (e.g., wall losses of particles) and also on some of the major assumptions (e.g., no losses of relatively volatile organic aerosol components and resulting evaporation in the line to the dilution chamber). More experimental details have been added together with an effort to better quantify the various uncertainties especially in the dilution measurements. We have also added material in the introduction to make sure that the appropriate credit is given to the publications that introduced the major ideas on which this work is based. These are described in more detail below in our responses to the specific comments of the referee.

**(2)** *In their introduction, Louvaris et al. completely downplay the similarity of their work to the previous work of Grieshop et al. (2009, EST). The title if the Grieshop paper is "Constraining the Volatility Distribution and Gas-Particle Partitioning of Combustion Aerosols Using Isothermal Dilution and Thermodenuder Measurements." This is extremely similar to the title of the current paper: "Estimation of the volatility distribution of organic aerosol combining thermodenuder and isothermal dilution measurements." In the introduction, the authors mention that "Volatility measurements [by Grieshop] based on longer equilibration timescales were performed for POA from a diesel engine and wood combustion using isothermal dilution." But, they do not even mention that Grieshop also performed thermodenuder measurements. This seems to me like inten-*

*tional obfuscation, especially since the authors also note that "Karnezi et al. (2014) proposed that volatility distribution of complex OA can be better constrained combining TD and isothermal dilution measurements," and later "we develop a new experimental technique to constrain the volatility distribution of organic aerosol using TD combined with isothermal dilution measurements following the suggestions of Karnezi et al. (2014)." Why mention Karnezi but not Grieshop in this context? Simply because one of the authors here is a co-author on the Karnezi paper but not the (earlier) Grieshop paper? I am concerned about this clear, seemingly intentional downplaying of previous measurements and methods.*

We never intended to downplay the contributions of the Grieshop et al. (2009) work. Please note that several of the authors of the present papers are colleagues and close collaborators of the authors of the Grieshop et al. (2009) work. We have rewritten the paragraph describing the efforts to combine dilution and thermodenuder measurements making sure that the Grieshop et al. study receives the credit that it deserves for suggesting and applying the main idea on which the Karnezi et al. (2014) work was based resulting in the next step in the present work.

**(3)** *The dilution experiments are insufficiently described. Was aerosol transferred once to the dilution chamber and then air sampled from it while the bag was allowed to collapse? Or was make up air continuously added? What was the flow rate through/out of the bag?*

The aerosol was transferred once to the dilution chamber and its evolution with time was followed. No additional air was added in these experiments, so the dilution ratio was constant with time. The sampling flow rate from the dilution chamber was 1 L $\text{min}^{-1}$. This information has been added to the revised paper.

**(4)** *Comparison of Fig. S3 and S4 indicates that for Experiment 2 the "corrected" number loss is 10-15 percent and, importantly, is very similar to the "corrected" mass loss.*

*In other words, comparison between these figures suggests, at least to me, that the mass loss is entirely driven by number loss for this experiment. I have similar concerns regarding the dilution data for experiment 1, from comparison between Fig. S2 and Fig. S6. Put another way, it seems that the number-normalized mass loss for the dilution experiments is close to zero (MFR close to 1). To be convincing that the mass loss is real, I think that the authors need to consider the extent to which the particle size distribution shifted. Was shrinkage observed to an extent that is consistent with 10-15 percent mass loss? Currently, I do not find the "corrected" mass loss experimental observations convincing. Consequently, I have concerns over the resulting model interpretation of the observations and the entire paper.*

This is a good point, as the dilution results are clearly sensitive to the accuracy of the wall loss corrections. We followed the suggestion of the reviewer and examined the aerosol number distributions to confirm that the particles were evaporating in the dilution chamber. In Experiment 1, the number mode diameter decreased from 93 nm to 87 nm corresponding to an 18 percent volume reduction. This is consistent with the 20 percent loss estimated after the wall loss correction. The shifting of the number distributions to smaller sizes was also obvious in the dilution data of Experiment 2. The corresponding change of the mode diameter was from 115 to 105 nm or a 24 percent reduction in volume. This information about the changes in the measured size distributions has been added to the manuscript to support the conclusions based on the loss-corrected measurements.

One important additional point can be made examining the estimated volatility distributions in Fig. 7. Based on the calculated uncertainty the mass fraction of the more volatile components (with C* higher or equal to 10 $\mu\text{g m}^{-3}$) could be as low as zero. This corresponds to practically no evaporation during dilution. As a result, our estimates considering their uncertainties are quite robust. On the contrary, the dilution data help constrain the upper limit of the contributions of the more volatile components to the COA.

**(5)** *L109: Since the authors are looking at "fresh" OA, it is unclear why OH radical concentrations are mentioned. D9 butanol is simply used as a dilution tracer.*

We have deleted this discussion of the calculation of OH concentrations that is probably confusing for most of the readers. There was a second chemical aging stage in the main smog chamber experiment, but this is not relevant for the present work that focuses on fresh OA.

**(6)** *L206 and Fig. 5: The experiments took nearly 8 h (30 mins at each of 15 temperatures). I think that the authors need to provide more information regarding exactly when the various spectra were measured. Is the "ambient" spectrum an average over those 8 h? Is it the average of the two measurements that came before/after the 200 C measurement? How similar in time was the ambient spectrum to the 200 C spectrum? Related, how similar were the ambient spectra measured at the beginning of the measurement versus 8 h later at the end of the measurement? Have things evolved over time due to the particles being suspended in the chamber? Granted, the spectra are quite similar, as the authors note, but more detail is required. Further, the authors simply note that the spectra are similar. But, their calculations suggest that, perhaps, they should be different because the particle composition at ambient vs. 200 C is different. This should be discussed.*

The spectrum of the ambient COA was practically constant during the experiment. The average spectrum is used in the paper. The theta angles between the spectra measured during the experiment and their average were all less than 2 degrees. The same stability characterized the spectra at 200 C. The angles between the individual measurements and the average spectrum were also less than 2 degrees. As a result, the comparison shown in Figure 5 is quite robust and applies throughout Experiment 1. This information about the stability of the measured mass spectra has been added to the revised manuscript.

The above results suggest that even if the cooking aerosol remaining at 200 C had quite

different volatility than the ambient COA, the differences of their AMS spectra were modest. This is probably due to the fact that the AMS measures mainly the fragments of the corresponding organic molecules which probably had a lot of similarities in this case. A brief discussion of this point has been added to the paper.

**(7)** *L217: The authors note that there are "small" differences between the two experiments. But I would argue that the differences, especially in the TD experiments, are actually quite large. I have included the figures from the paper below so that they can be directly compared. It is clear that the behavior at lower temperatures is dramatically different. One experiment indicates almost no evaporation until T > 80 C while the other indicates substantial (20 percent loss) evaporation at temperatures just over ambient. Further discussion is required.*

We agree with the referee that the use of the term "small differences" is probably confusing. We focus on the actual differences of the evaporated fraction at different temperatures and avoid qualifying them as small or large. Please note that some differences are expected given that these are two different cooking experiments with different meat, potential small differences in cooking details, etc.

**(8)** *L307 and elsewhere: The authors report the "average" volatility in a number of locations. First, it is not clear how this is calculated. Is it a linear average? A logarithmic average? Second, in reporting this number the authors seem to be making an a priori assumption that this is a meaningful number. This is not a value that is commonly reported. What is this mean meant to represent and how is it useful? I could have two very different distributions, for example one bimodal and one monomodal, that have the same average. These would exhibit very different behavior though.*

This is a logarithmic average, something that is now explained in the paper. We think that it is a useful metric of changes of a volatility distribution when the same volatility range (volatility bins) are used. We agree that in other cases it may not be as helpful

on its own, but here it expresses the change in the results due to the inclusion of the dilution measurements and we would prefer to keep it. In other cases, it could be accompanied by the standard deviation of the distribution to make it more useful.

---

## Author Comment (AC2) · 6 Aug 2017

**(1)** *This paper provided a new technique to measure/estimate the volatility distribution of organic aerosols by combining a thermodenuder (TD) and a dilution system. Cooking OA was used as an example to show the performance of this technique. In general, it could be a better technique than a separated TD system, which would be suffered by the possible processes occurred during the heating, e.g. thermal decomposition. The manuscript is overall well written and the topic fits the scope of AMT. I therefore recommend this manuscript can be published after some revision.*

We appreciate the positive assessment of our work by the referee. Detail responses to

the comments can be found below.

**(2)** *More discussions/statements are needed to explain how these two systems are combined. What is the role of dilution system in estimating the volatility distribution of targeted aerosol?*

The dilution data, at least in these experiments, help constrain the contributions of the more volatile organic aerosol components. These are the components with C* higher or equal to 10 $\mu$g m$^{-3}$ in this case (see Fig. 7). However, the dilution measurements cannot help constrain better the contributions of the LVOCs and ELVOCs to the organic aerosol composition. Even if the dilution results do not depend on the enthalpy of evaporation, their combination with the thermodenuder data does reduce the uncertainty of the corresponding estimate. These are now discussed in a new paragraph in the Conclusions section.

**(3)** *How to understand the differences between the modelled and experimented data during isothermal dilution (Fig. 4b)? It looks that the evaporation only occurred during the first few minutes, but not a continuous evaporation. Will this influence the understanding and estimation of the volatility distribution?*

This behaviour is quite useful in the estimation of both the volatility distribution of the cooking OA and the existence of any potential resistances to mass transfer. The prompt evaporation is interpreted by our model as evidence that any resistances to mass transfer in this system were modest. The resulting accommodation coefficients (evaporation coefficients in this case) were 0.06 and 0.07. The fact that only 20 percent of the COA evaporated during these first few minutes, suggests that the contribution of the more volatile OA components (IVOCs and part of the SVOCs) as also modest. After this 20 percent evaporated the system reached equilibrium and evaporation stopped. These useful insights from the behaviour of the dilution curve have been added to the paper.

[Figure]

**(4)** *It is highly possible that COA contain some ELVOC, e.g.  HULIS (Nie et al., 2017@ACP). I suggest the author to provide a more bin of ELVOCs?*

This is a good point that deserves a little more attention.  Almost all thermodenuder measurements stop at a temperature at which the MFR is not zero.  In our case at 250 C, the MFR was approximately 5 percent.  At least this fraction could in principle be ELVOCs.  Our model adds this fraction to the lowest volatility bin used ($C^*=10^{-3}$ $\mu$g m$^{-3}$).  In general, the contribution that appears in the lowest volatility bin refers to material that has volatility less or equal to the corresponding value.  This point is now made in the paper together with a reference to the work of Nie et al. (2017) discussing the contributions of HULIS to ELVOCs.

**(5)** *How to change the dilution time, by changing flow?*

There are a number of ways to increase the time available for dilution measurements. One is to take less frequent samples from the dilution chamber. Use of lower flow rates can also work. One could also use larger dilution chambers, which also help achieve higher dilution ratios, but are a lot more difficult to operate in the field.  Please note however that in these experiments the COA reached equilibrium after 30 min, so the remaining 2 hours of dilution data helped only in confirming the equilibrium state of the system.

**(6)** *Why the mass spectra look so similar before and after heating or dilution (25 C vs 200 C)? The volatility of organics should be correlated to their oxidation state (e.g. O/C), molecular weight et al. (E)LOVC tends to have a higher oxidation state and large molecular weight. This indicate there should be some differences for the mass fragmentation between evaporated mass and remained mass. PMF could be used for the AMS measurements in case of several heating steps, to make the change of mass spectra clearer, e.g. Hong et al., 2017@ACP.*

The modest difference in the AMS spectra of all the COA and its less volatile components is indeed interesting. This is probably due to the fact that the AMS measures mainly the fragments of the corresponding organic molecules which probably had a lot of similarities in this case. If the differences in volatility of compounds coming from the same source were mainly related to differences in the size of the organic molecules and not so much to their chemical nature (acids, hydrocarbons, etc.) one would expect such behaviour. Please note that there are differences (a theta angle of 11 degrees does support this) but that at least from the point of view of the AMS these differences are modest. PMF does not help in this case, because for two factors it results in more or less the ambient and the 200 C spectra that have already been analysed. A brief discussion of this point has been added to the paper.

**(7)** *Any ideas about the phase state of produced COA, which could influence the volatility measurement.*

These experiments do not provide much information about the phase state of the produced COA. The relatively fast evaporation during dilution does suggest relatively small delays to evaporation consistent with an evaporation coefficient of 0.06-0.07. These have already been considered in the analysis. In the sensitivity analysis section (Section 3.2.1) we discuss the sensitivity of our results to these uncertain mass transfer resistances considering the case of evaporation coefficients of 0.01 and 0.1. These tests suggested that the enthalpy of vaporization is rather insensitive to the evaporation coefficient in this range, and that the SVOCs and LVOCs changed by less than 15 percent. These are discussed in the Sensitivity Analysis section.

**(8)** *The residence time is effective residence time or total residence time? There would be a temperature profile in the thermodenuder? An effective residence time should be carefully considered.*

The residence time of 14 s mentioned in the paper is the centerline residence time at 298 K. The temperature profile (both in the longitudinal and radial directions) in our TD

has been analysed by Lee et al. (2010) and Gkatzelis et al. (2016). The change in volumetric flowrate due to the change in temperature along the TD is taken into account by the Riipinen et al. (2010) TD model used in this work. This information has been added to the manuscript.

**(9)** *The given vaporization enthalpy of COA in this study is a universal value, or can only be used in this work?*

The vaporization enthalpy of COA estimated here is applicable to pork meat char broiling. We do expect that it is a useful estimate for char broiling of other meat, but clearly this needs to be supported by other experiments. Its applicability to other types of cooking (e.g., frying) is unknown. We have added a qualifying statement about the use of the corresponding value.
* * *